# Synthesis of Network Biobased Aliphatic Polyesters Exhibiting Better Tensile Properties than the Linear Polymers by ADMET Polymerization in the Presence of Glycerol Tris(undec-10-enoate)

**DOI:** 10.3390/polym16040468

**Published:** 2024-02-07

**Authors:** Lance O’Hari P. Go, Mohamed Mehawed Abdellatif, Ryoji Makino, Daisuke Shimoyama, Seiji Higashi, Hiroshi Hirano, Kotohiro Nomura

**Affiliations:** 1Department of Chemistry, Tokyo Metropolitan University, 1-1 Minami Osawa, Hachioji, Tokyo 192-0397, Japan; lance.p.go@gmail.com (L.O.P.G.); mohamed-soliman@tmu.ac.jp (M.M.A.);; 2Osaka Research Institute of Industrial Science and Technology (ORIST), 1-6-50, Morinomiya, Joto-ku, Osaka 536-8553, Japan; higashi@orist.jp (S.H.);

**Keywords:** olefin metathesis, polymerization, biobased, isosorbide, polyesters, network, tensile property, chemical recycling

## Abstract

Development of biobased aliphatic polyesters with better mechanical (tensile) properties in film has attracted considerable attention. This report presents the synthesis of soluble network biobased aliphatic polyesters by acyclic diene metathesis (ADMET) polymerization of bis(undec-10-enyl)isosorbide diester [**M1**, dianhydro-*D*-glucityl bis(undec-10-enoate)] in the presence of a tri-arm crosslinker [**CL**, glycerol tris(undec-10-enoate)] using a ruthenium–carbene catalyst, and subsequent olefin hydrogenation using RhCl(PPh_3_)_3_. The resultant polymers, after hydrogenation (expressed as **HCP1**) and prepared in the presence of 1.0 mol% **CL**, showed better tensile properties than the linear polymer (**HP1**) with similar molecular weight [tensile strength (elongation at break): 20.8 MPa (282%) in **HP1** vs. 35.4 MPa (572%) in **HCP1**]. It turned out that the polymer films prepared by the addition of **CL** during the polymerization (expressed as a 2-step approach) showed better tensile properties. The resultant polymer film also shows better tensile properties than the conventional polyolefins such as linear high density polyethylene, polypropylene, and low density polyethylene.

## 1. Introduction

Development of biobased semicrystalline aliphatic polyesters derived from non-edible naturally abundant resources (plant oil, etc.) attracts considerable attention [1,2,3,4,5,6,7,8]. This is not only because these are considered alternatives to petroleum-based polymers but also because of their importance from the viewpoint of circular economy [9,10,11]. For example, these polyesters (including conventional polyesters) can be depolymerized by treating them with alcohols (via transesterification) in the presence of a catalyst to recover monomers exclusively (facile chemical recycling) [12,13,14,15]; one-pot closed-loop chemical recycling (depolymerization and repolymerization) was thus also demonstrated [13]. There have been thus many reports on the synthesis of polyesters derived from plant oil through polycondensation [7,15,16,17,18,19] and acyclic diene metathesis (ADMET) polymerization technique [7,17,20,21,22,23,24,25,26,27,28,29,30,31,32,33,34,35,36].

Recently, our laboratory demonstrated the synthesis of high molecular weight polyesters (expressed as **HP1** and **HP2**, Figure 1) [14,36] exhibiting better tensile properties (tensile strength, elongation at break) than conventional polyolefins, and the polymers (**HP1**) reported previously [20,21,32]. These polymers were prepared by ADMET polymerization [37,38,39,40] and subsequent hydrogenation. The effect of molecular weight on the tensile property was demonstrated by exhibiting better tensile properties [36].

In this paper, we focus on the synthesis of the soluble network polymers by ADMET polymerization. These polymers were prepared by conducting the polymerization in the presence of a crosslinker (**CL**) possessing three terminal olefins [31,41], as reported previously by us [31], and the resultant polymer should exhibit better tensile strength due to their network framework. We thus herein report the synthesis of network biobased aliphatic polyesters, which show better tensile properties (tensile strengths and elongation at breaks) than the linear one (**HP1**). These polymers were also depolymerized to afford the corresponding diesters and diols (and triols) through transesterification with ethanol in the presence of CpTiCl_3_ [12,14].

## 2. Materials and Methods

**General Procedure.** All synthetic experiments were conducted in a dry box or using standard Schlenk techniques under a nitrogen atmosphere. Anhydrous grade toluene, *n*-hexane, and dichloromethane (>99.5%, Kanto Chemical Co., Inc., Tokyo, Japan) were transferred into a bottle containing molecular sieves (mixture of 3A 1/16, 4A 1/8, and 13X 1/16) in a dry box. Isosorbide, 10-undecenoyl chloride, triethylamine, and glycerol of reagent grades (Tokyo Chemical Industry, Co., Ltd., Tokyo, Japan) were used without further purification. RuCl_2_(IMesH_2_)(CH-2-O*^i^*Pr-C_6_H_4_) [**HG2**; IMesH_2_ = 1,3-bis(2,4,6-trimethylphenyl)imidazolin-2-ylidene] obtained from Aldrich Chemical Co. (Milwaukee, WI, USA), was used as received. Monomer, dianhydro-*D*-glucityl bis(undec-10-enoate) (**M1**) was prepared according to the reported procedure [32].

All ^1^H and ^13^C NMR measurements were performed at 25 °C on a Bruker AV500 spectrometer (500.13 MHz and 125.77 MHz, respectively) using CDCl_3_ as solvent. Chemical shifts were reported as ppm with reference to SiMe_4_ at 0.00 ppm. Gel permeation chromatography (GPC) was used for the analysis of molecular weights and molecular weight distributions for the resultant polymer. The GPC measurements were carried out at 40 °C on a SCL-10A (Shimadzu Co., Ltd., Kyoto, Japan) connected columns (ShimPAC GPC-806, 804 and 802, 30 cm × 8.0 mm diameter, spherical porous gel made of styrene/divinylbenzene copolymer, ranging from <10^2^ to 2 × 10^7^ MW, Kyoto, Japan), using a Shimadzu RID-10A detector in THF (>99.8%, Kanto Chemical Co., Inc., Tokyo, Japan) served as the eluent with a flow rate 1.0 mL/min.

**Synthesis of crosslinker glycerol triundec-10-enoate (CL).** In the dry box, glycerol 300 mg, 3.25 mmol) and triethylamine (2.0 g, 19.8 mmol, 2.0 eq.) were added to THF (30 mL) and then cooled to 0 °C. 10-undecenoyl chloride (2.0 g, 9.9 mmol) was then added dropwise to the cooled glycerol solution. The reaction was allowed to react at room temperature overnight until completion by confirmation via TLC. The THF solvent was then removed with a rotary evaporator and then diluted with chloroform and washed with 2M HCl, 5% NaHCO_3_, deionized water, and brine. The washed product was then dried over MgSO_4_, filtered through a filter paper, evaporated using a rotary evaporator, and further dried under vacuum. The crude product was then purified using column chromatography (9:1, *n*-hexane: ethyl acetate), collecting a colorless oil of **CL** (1.54 g, 80% yield). The resultant product was further purified inside the dry box by dissolving the crosslinker in hexane and passing through a short column of alumina and celite. ^1^H NMR (CDCl_3_): δ 1.28 (br s, 24H, -C*H*_2_-), 1.37 (t, J = 6.9 Hz, 6H, -C*H*_2_-), 1.60 (m, 6H, -C*H*_2_CH_2_COO-), 2.03 (dt, J = 8.15, 7.4Hz, 6H, -C*H*_2_CH=CH_2_), 2.31 (t, J = 7.5 Hz, 6H, -C*H*_2_COO-), 4.14 (dd, J = 11.95, 5.97 Hz, 2H, -OC*H*_2_CHOCH_2_O-),4.29 (dd, J = 11.9, 4.31Hz, 2H, -OCH_2_CHOC*H*_2_O-), 4.92 (d, J = 10.21 Hz, 3H, C*H*_2_=CH-), 4.98 (d, J = 17.02 Hz, 3H, C*H*_2_=CH-), 5.26 (tt, J = 5.84, 4.35 Hz, 1H, -C*H*(CH_2_O)_2_-), 5.80 (ddt, J = 17.02, 10.26, 6.71 Hz, 3H, -C*H*=CH_2_) ^13^C{^1^H} NMR (CDCl_3_): δ 25.0 (-*C*H_2_-), 29.0 (-*C*H_2_-), 29.2 (-*C*H_2_-), 29.4 (-*C*H_2_-), 29.5 (-*C*H_2_-), 33.9 (-*C*H_2_CH=CH_2_), 34.2 (-*C*H_2_COO-), 34.4 (-*C*H_2_COO-), 62.3 (-*C*H_2_-OCO-), 69.0 (-*C*H-OCO-), 114.3 (*C*H_2_=CH-), 139.3 (-*C*H=CH_2_), 173.0 (-*C*OO-), 173.4 (-*C*OO-). APCI-MS: calculated for C_36_H_62_O_6_ *m/z,* 591.5 [M+H]^+^; found 591.4.

**ADMET Polymerization.** ADMET polymerizations were conducted by the analogous procedure reported previously [14,32]. In the dry box, the monomer (**M1**), crosslinker (**CL**), solvent, and ruthenium catalyst (**HG2**) were charged with the prescribed amounts in Table 1 into a sealed Schlenk-type tube (25 mL volume), and the reaction mixture was stirred for a specific time at 50 °C in an oil bath. The formed ethylene gas was removed continuously by freezing the reaction medium using liquid nitrogen, and the tube was shortly connected to the vacuum line with a certain time interval (each 15 min at the first 1 h, each 30 min in the following 2 h, and then each 1 h). The solvent exchange technique was conducted by solvent replacement [35,42] with a fresh solvent two or three times in the first two hours under N_2_ atmosphere. Two drops of EVE were used to quench the ADMET polymerization and stirring for 1 h. The reaction mixture was diluted using 4 mL chloroform and precipitated in 100 mL cold methanol. The resultant polymers were collected via filtration and dried in vacuo and were characterized using ^1^H (500.13 MHz) and ^13^C{^1^H} (125.77 MHz) NMR spectra in CDCl_3_ at 25 °C, GPC (SEC), and DSC.

**CP1**. ^1^H NMR (CDCl_3_): δ 1.28 (br s, 20H, -C*H*_2_-), 1.61 (tt, J = 14.66, 7.42 Hz, 4H, -C*H*_2_CH_2_COO-), 1.97 (m, 4H, -*CH*_2_CH=CH-), 2.30 (t, J = 7.46 Hz, 2H, -C*H*_2_COO-), 2.36 (t, J = 7.57 Hz, 2H, -C*H*_2_COO-), 3.79 (dd, J = 9.77, 5.45 Hz, 1H, -C*H*_2_CHO-), 3.93–4.00 (m, 3H, -C*H*_2_-CHO-, -C*H*_2_CHO-), 4.14 (dd, J = 12.01, 5.93 Hz, 2H, -C*H*_2_CHOCH_2_O-), 4.29 (dd, J = 11.76 Hz, 3.97 Hz, 2H, -C*H*_2_CHOCH_2_O-), 4.47 (d, J = 4.54 Hz, 1H, -C*H*-CHO), 4.82 (t, J = 4.92 Hz, 1H, -C*H*-CHO), 5.14 (dt, J = 5.60, 5.60 Hz, 1H, -C*H*OCO-), 5.19 (d, J = 2.80 Hz, 1H, -C*H*OCO-), 5.37 (m, 2H, -C*H*=C*H*-). ^13^C{^1^H} NMR (CDCl_3_): δ 25.0 (-*C*H_2_-), 29.2 (-*C*H_2_-), 29.6 (-*C*H_2_-), 29.4 (-*C*H_2_-), 29.8 (-*C*H_2_-), 32.7 (-*C*H_2_CH=CH-), 34.1 (-*C*H_2_COO-), 34.3 (-*C*H_2_COO-), 62.2 (-*C*H_2_-OCO-), 69.0 (-*C*H-OCO-), 70.5 (-*C*H_2_-CHO-), 73.6 (-*C*H_2_-CHO-), 73.9 (-*C*HOCO-), 78.0 (-*C*HOCO-), 80.9 (-*C*H-CHO-), 86.1 (-*C*H-CHO-), 130.0 (-*C*H=CH-), 130.5 (-CH=*C*H-), 173.0 (-*C*OO-), 173.3 (-*C*OO-).

**Olefin Hydrogenation.** The resultant polymers (200 mg), toluene (3.0 mL), and RhCl(PPh_3_)_3_ (3 mg) were added into the autoclave (20 mL scale) [36]. The stainless steel autoclave was then pressurized with hydrogen (1.0 MPa) and was placed into a heating Al block preheated at 50 °C. The solution was magnetically stirred for 24 h. The reaction mixture was then poured into a mixed solution of MeOH. The polymer precipitates were collected on a filter paper, and the collected white precipitates were then dried in vacuo for several hours. No significant differences in the *M*_n_, *Đ* (*M*_w_/*M*_n_) were observed in the polymer samples before/after the hydrogenation. The resultant polymers were identified by NMR spectra (shown in Appendix A), and the molecular weights and the distributions were analyzed by GPC. 

**HCP1**. ^1^H NMR (CDCl_3_): δ 1.27 (br s, 28H, -C*H*_2_-), 1.61 (tt, J = 14.75, 7.12 Hz, 4H, -C*H*_2_CH_2_COO-), 2.30 (t, J = 7.56 Hz, 2H, -C*H*_2_COO-), 2.36 (t, J = 7.51 Hz, 2H, -C*H*_2_COO-), 3.78 (dd, J = 9.95, 5.40 Hz, 1H, -C*H*_2_CHO-), 3.92–4.00 (m, 3H, -C*H*_2_-CHO-, -C*H*_2_CHO-), 4.14 (dd, J = 12.47, 5.92 Hz, 2H, -OC*H*_2_CHOCH_2_O-), 4.28 (dd, J = 11.77 Hz, 4.10 Hz, 2H, -OCH_2_CHOC*H*_2_O-), 4.47 (d, J = 4.55 Hz, 1H, -C*H*-CHO), 4.82 (t, J = 4.93 Hz, 1H, -C*H*-CHO), 5.14 (dt, J = 5.62, 5.55 Hz, 1H, -C*H*OCO-), 5.19 (d, J = 2.78 Hz, 1H, -C*H*OCO-). ^13^C{^1^H} NMR (CDCl_3_): δ 25.0 (-*C*H_2_-), 29.2 (-*C*H_2_-), 29.4 (-*C*H_2_-), 29.6 (-*C*H_2_-), 29.7 (-*C*H_2_-), 29.8 (-*C*H_2_-), 34.1 (-*C*H_2_COO-), 34.3 (-*C*H_2_COO-), 62.2 (-*C*H_2_-OCO-), 69.0 (-*C*H-OCO-), 70.5 (-*C*H_2_-CHO-), 73.6 (-*C*H_2_-CHO-), 73.9 (-*C*HOCO-), 78.0 (-*C*HOCO-), 80.9 (-*C*H-CHO-), 86.1 (-*C*H-CHO-), 173.0 (-*C*OO-), 173.3 (-*C*OO-).

**Analysis of Tensile Properties.** Stress/strain experiments were performed at 23 °C (speed 10 mm/min, humidity 50 ± 10%) using a Shimadzu Universal Testing Instrument (Autograph AGS-10kNX, Kyoto, Japan) equipped with load cell (cell capacity 500 N). At least three specimens were tested for analysis of each polymer. The small dumbbell-shaped test specimens [width of parallel portion of 3 mm, distance between grippers (or grippering distance) of 10 mm, overall length of 25 mm and thickness of 0.1 mm] were cut from the polymer sheets prepared by a hot press. A toluene solution (2.0 mL) of **HP1** or **HCP1** (200 mg) sonicated for 10 min was placed into a PTFE petri dish (φ50 mm) and the sample films were prepared by removing the solvent upon heating (on a hot plate). The resultant solvent casted films were partly placed into a compression molding press machine (Shinto Metal Industries, Ltd., Osaka, Japan), and steel plates were heated up to 100 °C under 5.0 MPa for 2 min. The films were then obtained after cooling the steel plates to room temperature.

**Scheme 2 polymers-16-00468-sch002:**
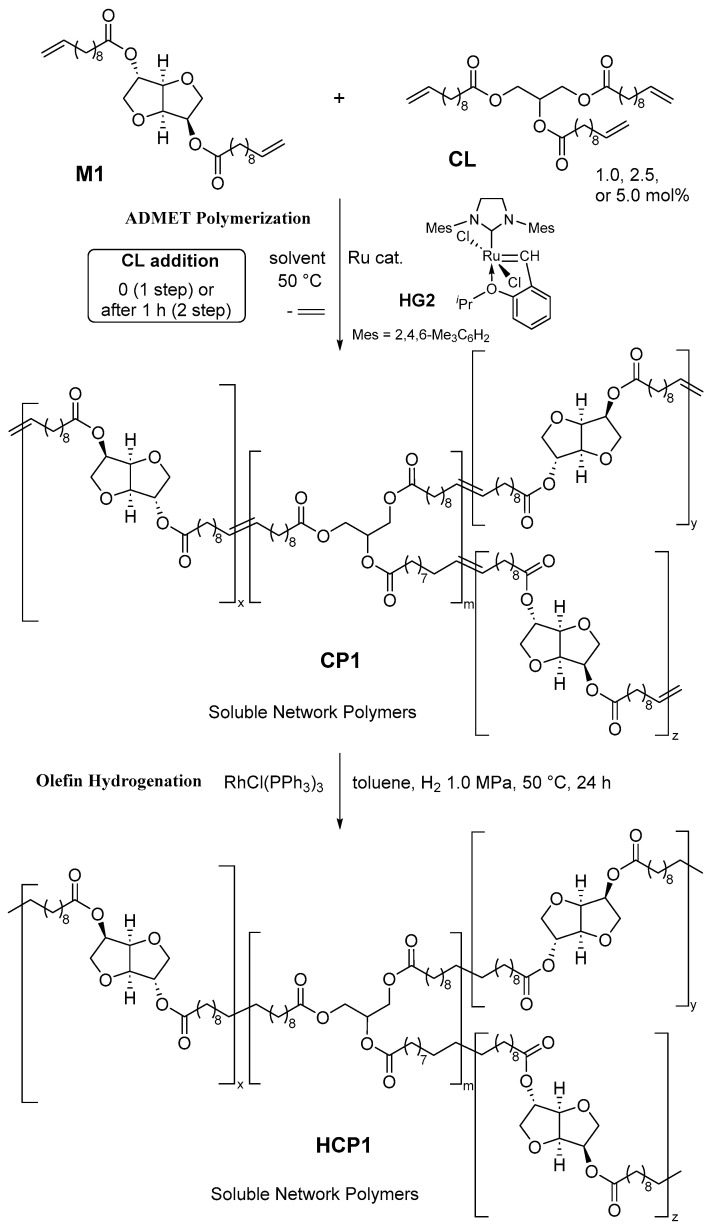
Synthesis of network polymers (**CP1** and **HCP1**) by ADMET polymerization and subsequent hydrogenation.

## 3. Results and Discussion

### 3.1. Synthesis of Network Polymers (CP1 and HCP1) by ADMET Polymerization and Subsequent Hydrogenation

ADMET polymerization of bis(undec-10-enyl)isosorbide diester [**M1**, bis(undec-10-enoate) with isosorbide] in the presence of glycerol tris(undec-10-enoate) (**CL**) was chosen because the resultant polymer film prepared by **M1** (expressed as **HP1**, Figure 1) exhibited good tensile strength and elongation at break [36]. As shown in Figure 2, two approaches involving adding **CL** from the beginning (1-step approach) or after 1 or 3 h (2-step approach) have been chosen for the synthesis since the approach might affect the network density or average polymer chain length between **CL**s (crosslinking point). The polymerizations of **M1** by the ruthenium–carbene catalyst (**HG2**, 2.0 mol%) were conducted in solvent (toluene, chloroform, or tetrachloroethane, initial **M1** conc. 0.94 mmol/mL) in the presence of **CL** (0.5–5.0 mol%), which were prepared by glycerol with 3.0 equiv of 1-undecenoyl chloride (see Materials and Methods). The selected results conducted under the various conditions are summarized in Table 1. For the obtainment of high molecular weight polymers under these conditions (300 mg **M1** scale), the solvent in the reaction mixture was removed in vacuo to replace the fresh one (called solvent replacement) every 30 min in certain experimental runs [42]. The method is effective for the purpose (removal of ethylene remained) of condensation polymerization [42], although the method would not be appropriate in terms of a green sustainable process, and alternative methods (conducted in IL [14] or a molybdenum catalyst [36]) should be studied.

It was revealed that the ADMET polymerizations of **M1** in the presence of 0.5–2.5 mol% **CL** (added at the beginning, called a 1-step approach) gave the highest molecular weight polymers (expressed as **CP1**s), and the *M*_n_ values became higher upon presence of **CL** (0.5 or 1.0 mol%, runs 1–5) compared to that in the absence of **CL** (run 1). The *M*_n_ value decreased upon further **CL** addition (2.5 mol%, runs 9 and 10), and the resultant polymers became swelled gels with stop stirring when the polymerizations were conducted in the presence of 5.0 mol% of **CL** (run 14). The results are highly reproducible, as demonstrated in runs 3–5, although the dispersity (PDI, *M*_w_/*M*_n_) values are somewhat large, probably due to the difficulty of stirring the reaction mixture owing to high viscosity (run 14) [14]. Indeed, the polymerizations in the presence of 5.0 mol% **CL** eventually afforded polymer gels irrespective of the kind of solvent employed (runs 13–17). It seems that the polymerization conducted in toluene and tetrachloroethane afforded polymers possessing rather low molecular weights compared to those conducted in chloroform, whereas the solvent replacement improved the *M*_n_ value in **CP1** [*M*_n_ = 19,300 (run 6) vs. 26,000 (run 7)].

In contrast, the *M*_n_ values in the resultant polymers (**CP1**s) were low when **CL** (1.0 mol%) was added after 1 h of the ADMET polymerization (expressed as a 2-step approach) irrespective of solvents (*M*_n_ = 17,600–26,800, runs 18–20); the polymerization in chloroform gave **CP1** with the highest molecular weight (run 18). Although the *M*_w_/*M*_n_ values in the resultant polymers in the presence of 2.5 mol% **CL** conducted in chloroform were rather large [*M*_w_/*M*_n_ = 4.13–5.45 (runs 23–25)], the *M*_w_/*M*_n_ values in **CP1** became rather low (unimodal) when **CL** was added after 3 h of the ADMET polymerization (runs 29–32). As observed above, an increase in the number of solvent replacements led to an increase in the *M*_n_ values because the ethylene that remained in the polymerization solution was removed in vacuo with the removal of solvent, which led to condensation polymerization [42]. Polymer samples with different molecular weights for the analysis of tensile properties (shown below, Figure 1 and Figure 2) were thus prepared by adopting this method.

According to the reported procedure [14], olefinic double bonds in the resultant polymers (**CP1**s) were hydrogenated by RhCl(PPh_3_)_3_ catalysts in toluene (Figure 2, H_2_ 1.0 MPa, 50 °C, 24 h). The results are summarized in Table 2. It was revealed that, as reported previously [36], no significant changes in the *M*_n_ (as well as *M*_w_/*M*_n_) values were seen before/after hydrogenation, and resonances ascribed to the internal olefins disappeared in the polymer samples (**HCP1**s) after hydrogenation. Their uniform compositions (completion of hydrogenation) were also confirmed by DSC thermograms observed as sole melting temperatures [32]. As shown in Appendix A (and Appendix A), no significant differences in the melting temperature were observed between linear and network polymers, whereas the *T*_m_ values in **HCP1** are higher than **CP1,** as reported previously by **P1** and **HP1** [32]. The resultant polymers are soluble in chloroform, THF, and toluene, regardless of their network structure.

### 3.2. Tensile Properties in the Polymer Films (CP1s, HCP1s)

Small dumbbell-shaped test specimens in the resultant polymer samples were prepared by cutting the polymer sheet for measurement of their tensile properties. The polymer sheets were prepared using a hot press method according to the reported procedure [36]. The stress/strain experiments were conducted using a universal testing machine at 23 °C (speed of 10 mm/min, humidity 50 ± 10%). The selected results are shown in Figure 1 and Figure 2, and the data are summarized in Table 3.

It should be noted that, as shown in Figure 1a, the tensile strength (stress) in the resultant network polymer films after hydrogenation (**HCP1**) became higher than that prepared in the absence of **CL** (**HP1**, linear polymer). The elongation at break (strain) was affected by the method prepared, whereas no significant effects toward the stress were observed; the polymer film prepared by the 2-step approach (addition of **CL** after 1 h polymerization) showed higher strain compared to that prepared by the 1-step approach. This might be probably due to the difference in the average length of each polymer chain between **CL** units, although we do not have clear evidence at this moment. Increased **CL** (from 1.0 mol to 2.5 mol%) led to a decrease in the strain, probably due to increased network density. As reported previously [36], both the tensile strength and the elongation break in the linear polymer (**HP1**) are affected by the *M*_n_ value [**HP1**, *M*_n_ = 30,700 vs. 40,900. Figure 1b]. In contrast, in the network polymer (**HCP1**), an increase in the tensile strength (strain) was observed upon an increase in the *M*_n_ value [**HCP1**, *M*_n_ = 31,400 vs. 37,300. Figure 1b].

Figure 2 shows tensile properties in the resultant unsaturated polyester films (**CP1**, before hydrogenation) prepared by the ADMET polymerization. It was also noted that both the tensile strength (stress) and the elongation at break in the resultant network polymer films (**CP1**) became higher than that prepared in the absence of **CL** (**P1**, linear polymer). As reported previously in **HP1** [36], the resultant polymer films (**CP1**) showed higher stress (elongation at break) than the saturated ones (**HCP1**), although the tensile strengths (strain) were somewhat low. In contrast to the results in the saturated polymer films (**HCP1**), it seems that the tensile properties were not affected by the method employed (1-step or 2-step), whereas the strain (elongation at break) in **CP1** decreased upon increasing the **CL** (from 1.0 mol to 2.5 mol%).

Figure 3 summarizes plots of tensile (fracture) strengths and strains (elongation at breaks) of **HP1** and **CHP1** for comparison with the conventional polymers such as linear high density polyethylene (HDPE), polypropylene (PP) and low density polyethylene (LDPE) [43]. It is clear that the resultant polymer film shows better tensile properties (tensile strength, elongation at break) than the conventional polyolefins as well as the other conventional polymers.

Figure 4 shows temperature dependence in the storage module (*E*′), loss module (*E*″), and loss factor, tan *δ*, for **HP1** and **HCP1**s measured by dynamic mechanical analyses (DMA, 1.0 Hz). The resultant polymers possessed tan d values with relatively narrow widths, suggesting that the resultant polymer possessed uniform composition, and the results may also suggest that the crosslinking distributions in **HCP1** were uniform [44]. However, no significant differences in their temperature dependences toward *E*′, *E*″, and tan δ were observed between linear (**HP1**) and network polymers (**HCP1**). It seems likely that the resultant polymers possessed relatively low crosslinking density, and polymer units between the crosslinking points are relatively long (reflect the polymer property measured by DMA analysis).

## 4. Conclusions

The present report demonstrates synthesis of soluble network biobased aliphatic polyesters by ADMET polymerization of bis(undec-10-enoate) with isosorbide [**M1**, bis(undec-10-enyl)isosorbide diester] in the presence of tri-arm crosslinker (**CL**), glycerol tris(undec-10-enoate). The resultant polymers, after hydrogenation (expressed as **HCP1**) and prepared in the presence of 1.0 mol% **CL**, showed better tensile properties than the linear polymer (**HP1**) with similar molecular weight. The addition of CL during the polymerization (expressed as a 2-step approach) was effective for the preparation of polymer films with better tensile properties. As shown in Figure 3, the resultant polymer film shows better tensile properties than the conventional polyolefins such as linear high density polyethylene (HDPE), polypropylene (PP), and low density polyethylene (LDPE) as well as the other conventional polymers. The approach should be effective for the synthesis of other biobased polyesters to improve especially the elongation at break.

Moreover, as demonstrated in **HP1** [14], the resultant polymers (**HCP1**s) could be depolymerized with ethanol in the presence of CpTiCl_3_ (1.0 mol%, 150 °C, 24 h) to afford isosorbide and the dicarboxylic acids confirmed by NMR spectra (details are shown in the Appendix A). Therefore, these polyesters could demonstrate a promising possibility of chemical recyclable, biobased aliphatic polyesters not only as alternatives to conventional polyolefins but also as functional polymers suited to a circular economy.

## Data Availability

Data is contained within the article and the Appendix A.

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
