# Peer review of "Synthesis of Network Biobased Aliphatic Polyesters Exhibiting Better Tensile Properties than the Linear Polymers by ADMET Polymerization in the Presence of Glycerol Tris(undec-10-enoate)"

_polymers, 2024, doi:10.3390/polym16040468_

Round 1

Reviewer 1 Report

Comments and Suggestions for Authors

Nomura and coauthors have reported a new high-strength polymeric network crosslinked with tri-arm crosslinker [CL, tris(undec-10-enoate), which is much better than common reported polymeric networks. This work has relatively high originality and interesting.

However, a major revision is needed before acceptation. Here are some suggestions and comments for this research work to further enhance this work below:

1. The title should be shortened and revised to highlight the innovation better.

2. The abstract part just stated the synthesis and performance of this new materials, which should be a “story”. I suggest authors can further illustrate the advantage of this polymeric network and rearrange the logic of this “story” including “Why and How” to do this work and the comparison of mechanical properties with other linear polymers.

3. The introduction part are too short to clear expound the research background and existing problems of this project related to this work.

4. The data to supporting the conclusion of this work are not enough. Some key data (such as the comparison of SEM images, DSC, TGA and so on between this materials and other linear polymers) should be added.

5. Authors state the solubility of this polymers is depended on the added percentage of the tri-arm crosslinker, which should be further explained to illustrate the mechanism and added some references.

6. Figure 4 in the conclusion part should be placed in the “result and discussion”

7. The conclusion part should be drastically revised to highlight the advantage of this polymer material.

Comments on the Quality of English Language

English language is relatively good, which just need minor revised.

Author Response

Dear reviewer 1:

Thank you for reviewing the manuscript.  We have thus revised carefully according to your comments and the comments from the other reviewers.  The point-by-point responses to your comments are as follows.

  1. The title should be shortened and revised to highlight the innovation better.

Thank you for your comment.  We also received the similar comment and have revised (simplified) the title.  We highly believe this could fulfill your request.

  1. The abstract part just stated the synthesis and performance of this new materials, which should be a “story”. I suggest authors can further illustrate the advantage of this polymeric network and rearrange the logic of this “story” including “Why and How” to do this work and the comparison of mechanical properties with other linear polymers.

Thank you for the helpful comment.  We have thus revised the abstract according to your comment.  We believe the revision could fulfill your request.

  1. The introduction part are too short to clear expound the research background and existing problems of this project related to this work.

Thank you for your comment and we have revised the introductory part on this occasion.  We highly believe that this could be fine, since the introductory part should not be too long but should be consistent with the purpose on this subject.

  1. The data to supporting the conclusion of this work are not enough. Some key data (such as the comparison of SEM images, DSC, TGA and so on between this materials and other linear polymers) should be added.

Thank you for your comment and we have thus placed the DSC thermograms in the Supporting Information.  We do not see significant differences between the linear polymer and the network polymers.  We believe that this could be fine for your request.

  1. Authors state the solubility of this polymers is depended on the added percentage of the tri-arm crosslinker, which should be further explained to illustrate the mechanism and added some references.

Thank you for your critical comment.  We feel sorry that we have revised the expression as follow, because the reaction mixture became gel with stop stirring not became insoluble.  We highly hope that the revision could be fine for your comment.  We hope that the reviewer could kindly understand this revision.

… the resultant polymers became insoluble swelled gel with stop stirring, when the polymerizations were conducted in the presence of 5.0 mol% of CL…

  1. Figure 4 in the conclusion part should be placed in the “result and discussion”
  2. The conclusion part should be drastically revised to highlight the advantage of this polymer material.

Thank you for the comment and we have placed it before Figure 3 and the changed the order of the references.  We hope this could be fine for your request.

We have also revised the concluding remarks upon your comments to place more words to summarize the content including emphasis.  We highly hope that the revision could fulfill your request.

We have also revised the manuscript according to the comments from the other reviewers.  We highly hope that these revisions could fulfill your requests.  We highly believe that the manuscript will be accepted and introduced through your journal in the near future.

Yours sincerely, 

Kotohiro Nomura, Tokyo Metropolitan Universioty, Japan

Reviewer 2 Report

Comments and Suggestions for Authors

This is an interesting manuscript concerning fundamental studies of ADMET polymerization of isosorbide bis(undec-10-enoate) and its crosslinking with different amounts of tri-arm crosslinker (CL), glycerol tris(undec-10-enoate).

1. A title of the manuscript should be corrected, for example as follows:

"Synthesis of Soluble Biobased Branched Aliphatic Polyesters Exhibiting Better Tensile Properties than the Linear Polymers obtained by ADMET Polymerization of Isosorbide Bis(undec-10-enoate) in the Presence of Glycerol Tris(undec-10-enoate)".

2. What is a difference in chemical structures of monomers M1 and M2 (see Scheme 1) ? Was a mixture of M1 and M2 used ?

3. M1 is not the "isosorbide", but it is "bisundecanoyl isosorbide diester".

4. I suggest to change a term "network polymers" for "branched polymers".

5. Values of a tensile strength and an elongation at break for HDPE, LDPE and PP should be added.

6. A name of tri-arm crosslinker (CL), "tris(undec-10-enoate) with glycerol" should be changed for "glycerol tris(undec-10-enoate)". 

7. Taking into consideration experimental conditions of dry box and Schlenk techniques and quite complex experimental procedure and also a cost of catalysts the studied ADMET polymerization method will not find a practical application for a long time.

8. A removal of the catalysts from prepared polyesters would be rather difficult and expensive.

9. I have doubts if the proposed recycling method of prepared polyesters with ethanol has a sense from a practical point of view ? How Authors plan to reuse isosorbide and diester prepared in the recycling process ?

Author Response

Dear reviewer 2:

Thank you for reviewing the manuscript.  We have thus revised carefully according to your comments and the comments from the other reviewers.  The point-by-point responses to your comments are as follows.

  1. A title of the manuscript should be corrected, for example as follows:

"Synthesis of Soluble Biobased Branched Aliphatic Polyesters Exhibiting Better Tensile Properties than the Linear Polymers obtained by ADMET Polymerization of Isosorbide Bis(undec-10-enoate) in the Presence of Glycerol Tris(undec-10-enoate)".

Thank you for the comment.  Since we received the comment of the revision of the title, we simplified it on this occasion.  As we will submit another paper that the resultant network polymer showed promising capabilities for preparation of semi-IPN exhibiting that further improves mechanical property.  We highly believe that this could be fine for your request.

  1. What is a difference in chemical structures of monomers M1and M2(see Scheme 1) ? Was a mixture of M1 and M2 used ?

The monomer we employed was M1 (isosorbide based) not M2 (isomannide based), because both showed similar mechanical properties in film, as reported previously by us (ACS Macro Lett. 2023, 12, 1403-1408.).  As you can find, M1 and M2 are diastereomer.

  1. M1is not the "isosorbide", but it is "bisundecanoyl isosorbide diester".

Thank you for the comment and we have revised.

  1. I suggest to change a term "network polymers" for "branched polymers".

Thank you for the critical comment.  We feel very sorry that we will use the term “network polymer” because we do not see any residual terminal olefinic double bonds in the resultant polymers, clearly suggesting the formation of certain network.  As described above, we recently succeeded preparation of semi-IPN with cellulose nanofibers (CNF) that further improves mechanical properties, whereas simple mixing the liner polymer and CNF does not show such properties.  Moreover, as cited in reference 24 (RSC Adv. 2019, 9, 10245-10252.), we previously reported synthesis of network polymers by adopting the ADMET polymerization in the presence of tri-arm undecenoate.  We highly hope that the reviewer kindly understands this point.

  1. Values of a tensile strength and an elongation at break for HDPE, LDPE and PP should be added.

Thank you for the comment and we have placed the data in the figure caption (Figure 3 in the revised manuscript).  We highly believe that this could fulfill your request.

  1. A name of tri-arm crosslinker (CL), "tris(undec-10-enoate) with glycerol" should be changed for "glycerol tris(undec-10-enoate)". 

Thank you for the comment and we have revised.

  1. Taking into consideration experimental conditions of dry box and Schlenk techniques and quite complex experimental procedure and also a cost of catalysts the studied ADMET polymerization method will not find a practical application for a long time.
  2. A removal of the catalysts from prepared polyesters would be rather difficult and expensive.

Thank you for your critical comment.  As you can understand this is a basic polymer science manuscript and can be modified for the practical application.  I strongly (KN) believe this could be fine as the basic polymer science paper, because we need to introduce scientifically new in the academic manuscript.

As commented, complete removal of ruthenium catalyst is somewhat difficult as introduced in some papers.  However, as you may know, some companies already demonstrated this matter with commercial level.  We also have the technique for the removal. 

  1. I have doubts if the proposed recycling method of prepared polyesters with ethanol has a sense from a practical point of view ? How Authors plan to reuse isosorbide and diester prepared in the recycling process ?

As introduced in the previous paper, the resultant polymer can be depolymerized to give diesters and isosorbide, that can be separated simply by precipitation.  The comment is out of discussion for this paper but we will introduce the results of closed loop chemical recycling and the upcycling of the polymers in the very near future.  We have the collaborative project with industry, since the process is very simple and applicable.

We have also revised the manuscript according to the comments from the other reviewers.  We highly hope that these revisions could fulfill your requests.  We highly believe that the manuscript will be accepted and introduced through your journal in the near future.

Yours sincerely, 

Kotohiro Nomura, Tokyo Metropolitan Universioty, Japan

Reviewer 3 Report

Comments and Suggestions for Authors

Journal: Polymers

Manuscript ID: 2826608

Title: Synthesis of Soluble Network Biobased Aliphatic Polyesters Exhibiting Better Tensile Properties than the Linear Polymers by ADMET Polymerization of Bis(undec-10-enoate) with Isosorbide in the Presence of Tris(undec-10-enoate) with Glycerol.

The research presents the synthesis of soluble network biobased aliphatic polyesters through acyclic diene metathesis (ADMET) polymerization. This study builds upon the author's previous work (ACS Omega 2023, 8, 7222-7233 and ACS Macro Lett. 2023, 12, 1403-1408) and involves a meticulous exploration of polymer synthesis and structural characterization. The resulting network polymers exhibit superior tensile properties compared to linear polymers with similar molecular weights. The manuscript is recommended for publication after revisions.

The feedback and recommendations are as follows:

1.     The manuscript should adhere to the updated IUPAC nomenclature, replacing "PDI" (polydispersity index) with "dispersity."

2.     It is suggested to label all peaks in the NMR spectra in the Supplementary Information (SI).

3.     Including information on the thermal and crystalline properties of best performing network polymers would enhance the manuscript quality.

4.     Shortening the title of the manuscript is recommended.

5.     Although depolymerization data is included in the SI, there is no discussion in the manuscript or SI. It is suggested to discuss depolymerization in the manuscript.

Author Response

Dear reviewer 3:

Thank you for reviewing the manuscript.  We have thus revised carefully according to your comments and the comments from the other reviewers.  The point-by-point responses to your comments are as follows.

  1. The manuscript should adhere to the updated IUPAC nomenclature, replacing "PDI" (polydispersity index) with "dispersity."

Thank you for the comment and we have revised according to your comment.  We hope the revision could fulfill your request.

  1. It is suggested to label all peaks in the NMR spectra in the Supplementary Information (SI).

Thank you for the comment and we have revised the NMR charts with assignment.  We hope this could be fine for your request.

  1. Including information on the thermal and crystalline properties of best performing network polymers would enhance the manuscript quality.

Thank you for your critical comment and we have thus added the DSC thermograms.  We hope this could be fine for your request.

  1. Shortening the title of the manuscript is recommended.

Thank you for the comment.  Since we received the similar comment from the other reviewers, and we have thus revised (simplified) the title in this manuscript.  We hope this could fulfill your request.

  1. Although depolymerization data is included in the SI, there is no discussion in the manuscript or SI. It is suggested to discuss depolymerization in the manuscript.

Thank you for your comment.  Indeed, we placed the fact in the text (concluding remarks) in the submitted manuscript.  For better explanation, we have placed a note in the revised manuscript.  We hope the revision could fulfill your request.

We have also revised the manuscript according to the comments from the other reviewers.  We highly hope that these revisions could fulfill your requests.  We highly believe that the manuscript will be accepted and introduced through your journal in the near future.

Yours sincerely, 

Kotohiro Nomura, Tokyo Metropolitan Universioty, Japan

Reviewer 4 Report

Comments and Suggestions for Authors

In this manuscript, Nomura et al. reported the synthesis of the network biobased aliphatic polyesters via ADMET polymerization. The as-obtained network polymers show better tensile properties than the linear analog. Overall, the manuscript is well-written. I suggest the authors add some discussion about the properties and advantages of network polymers.

Comments on the Quality of English Language

Moderate editing is needed

Author Response

Dear reviewer 4:

Thank you so much for reviwing the manuscript.  We have revised the manuscript according to comments from the other reviers and we hope this should be fine.

Yours sincerely,

Kotohiro Nomura, Tokyo Metropolitan Uiversity, Japan

Round 2

Reviewer 1 Report

Comments and Suggestions for Authors

This revised manuscript can be accepted without any reversion.